

# Identification of chromosome ploidy and karyotype analysis of cherries (*Prunus pseudocerasus* Lindl.) in Guizhou

Nian Chen*, Yali Wang*, Mei He, Fei An, Jiyue Wang and Changmei Song

Key Laboratory of Surveillance and Management of Invasive Alien Species in Guizhou Education Department, Platform for Exploitation and Utilization of Characteristic Plant Resources, College of Biological and Environmental Engineering, Guiyang University, Guiyang, Guizhou, China
* These authors contributed equally to this work.

## ABSTRACT

The present study aimed to characterize the chromosome features of cherry (*Prunus pseudocerasus* Lindl.) germplasm in Guizhou Province, China, in order to facilitate the selection and breeding programs of this economically and ornamentally valuable species. The stem tip chromosome preparation technique was employed for ploidy identification and karyotype analysis, and the results were further validated by flow cytometry. The findings revealed that the 28 cherry accessions from Guizhou province exhibited a chromosomal base of x = 8, comprising 19 tetraploid and nine hexaploid individuals. Karyotype analysis showed two chromosome types, "m" and "sm," with the longest chromosome/shortest chromosome (Lc/Sc) ranging from 1.6 to 2.65, the mean arm ratio (MAR) varying from 1.15 to 1.56, and index of the karyotypic asymmetry (As.K) ranging from 53.74 to 61.6. Three karyotype types, "1A," "1B," and "2B," were identified among the studied accessions. The most evolutionarily advanced accession was HZ152, while DCZC27 represented the most primitive karyotype. This study expands the ploidy database of Chinese cherry and provides valuable information for the conservation and utilization of cherry germplasm resources in Guizhou province.

## INTRODUCTION

The *Prunus pseudocerasus* Lindl., belonging to the genus Prunus within the Rosaceae family, is primarily distributed across the northern temperate zone, with China hosting the largest number of species in this genus (*Wang et al., 2023a*). *Cerasus* plants are naturally found in temperate regions of Asia, Europe, and North America (*Zhang et al., 2021*). One such species, the Chinese cherry (*Prunus pseudocerasus* Lindl.), is widely distributed in the subtropical and temperate regions of China (*Yi et al., 2020*). Guizhou Province, situated in southwestern China, is renowned for its expansive karst landscapes, which constitute one of the world's largest continuous karst systems (*Long et al., 2021*). The *Prunus* species

Corresponding author
Changmei Song, gzgyscm@126.com

found in Guizhou include saucer *Prunus* and *Pseudoprunus* (*Xu et al., 2023*). Cherries are an abundant fruit resource and serve as an excellent gene pool for variety breeding.

Chromosome number and karyotype analysis are essential tools for both qualitative and quantitative assessments of plant chromosomal characteristics. Previous research has established that polyploidy significantly contributes to the evolutionary processes of eukaryotes, resulting in organisms characterized by more than two complete sets of chromosomes (*Van de Peer, Mizrachi & Marchal, 2017*). In plants, various mechanisms such as mutation, recombination, selection, and isolation play essential roles in species evolution, particularly among flowering plants (*De Bodt, Maere & Van de Peer, 2005*). Polyploidy facilitates chromosomal rearrangements that enhance individual adaptation (*Dunham et al., 2002*; *Gerstein et al., 2006*). Additionally, karyotype variability and chromosomal polymorphism are commonly observed across different plant species (*Warchałowska-Śliwa et al., 2021*). Notably, *Liu et al. (2010)* found that the karyotype asymmetry coefficient and the average arm ratio exert a more significant influence on evolutionary outcomes than the chromosome length ratio. Furthermore, *Yang et al. (2024)* posited that a higher karyotype asymmetry coefficient is indicative of greater evolutionary advancement within the plant kingdom. Multivariate quantitative approaches have been suggested for inferring different karyotypic parameters, such as chromosome number, basic chromosome number, total haploid chromosome length, and mean centromeric asymmetry (*Peruzzi & Altınordu, 2014*). Flow cytometry is a powerful technique for chromosome analysis, enabling the study of chromosome number, structure, and DNA content. *Bartholdi et al. (1987)* described methods for chromosome sorting by flow cytometry, emphasizing the importance of high purity and optimal rates. Additionally, a procedure for the preparation of intact mitotic chromosomes from seedlings for flow cytometric analysis has been developed (*Doležel et al., 2011*). Protocols for separating and staining metaphase chromosomes have also been established to prepare single-chromosome suspensions for flow cytometry and sorting (*Mukhopadhyay et al., 2023*). The karyotype of the *Prunus pseudocerasus* Lindl. is highly valuable for auxiliary breeding, as studying and understanding the genetic makeup of these plants can provide valuable insights for genetic improvement and conservation programs (*Pinosio et al., 2020*). Determining the ploidy of Chinese cherry germplasm in Guizhou Province is crucial for breeding, analyzing genetic relationships among germplasm resources, effectively utilizing regional germplasm, and understanding the ploidy and karyotype characteristics of cherry and its relatives.

*Wang et al. (2018)* used representative wild and cultivated samples from sixteen natural populations in four provinces (Sichuan, He'nan, Shaanxi and Chongqing) of China to study the ploidy level of Chinese cherries, the sixteen Chinese cherry populations were predominantly tetraploid with 2n = 4x = 32, except for two populations, pentaploid and hexaploid; and *Wang et al. (2023b)* analyzed the ploidy levels in self-and open-pollinated seedling progenies of tetraploid and hexaploid Chinese cherry, their result showed that both self- and open-pollinated progenies of tetraploid Chinese cherry exhibited tetraploids, pentaploids, and hexaploids, with tetraploids being the most predominant. In the seedling progenies of hexaploidy Chinese cherry, the majority of hexaploids and a few pentaploids

were observed. But its abundant germplasm resources, dispersed distribution and complex ploidy levels (*Gepts, 2006*), the current results are few and require further analysis. To address this issue, the present study aimed to determine the chromosome ploidy of 28 cherry germplasm resources from Guizhou Province, China. Chromosome counting and flow cytometry analysis were employed to identify the ploidy levels of these resources. Furthermore, karyotype analysis and phylogenetic analysis were performed to provide genetic and cytological insights for resource identification, protection, and utilization, as well as to generate necessary cytological data for future breeding efforts.

## MATERIALS AND METHODS

The materials for this experiment were sourced from Chinese cherry germplasm resources in different regions of Guizhou, with specific collection sites detailed in Table 1.

### Chromosome preparation

The stem tips were gathered one sunny morning (9:00–11:00). The outer layer of young leaves on the stem tips was carefully peeled off, leaving a length of 0.5–1 cm from the base. Immerse the stem tip in a 1:1 mixture of V (0.002 mol/L octa-hydroxyquinoline) and V (saturated p-dichlorobenzene), or V (0.002 mol/L octa-hydroxyquinoline) and V (0.02% colchicine) for 3 h of light treatment at 4 °C. Subsequently, thoroughly wash the pre-treated stem tips with deionized water before transferring them to V (anhydrous ethanol):V (glacial acetic acid) = 3:1 Carnot fixing solution. Fix the tips at 4 °C for 12–24 h. After cleaning, temporarily store the deionized water in 70% alcohol for future use. Following 3–5 washes with distilled water, the pre-treated stem tips were treated with a 0.5 mol/L HCl aqueous solution at 60 °C for 15 min to dissociate them. Any residual hydrochloric acid was then thoroughly rinsed off using deionized water. Stain the tips of the detached stems with modified phenol fuchsin dye solution for 40 min. Place the stained tips on a slide and use a dissection needle under a stereomicroscope to remove the base. Next, carefully select the well-stained tips, press down the cover glass, and gently tap the back of the blade locally to disperse the cells. Afterwards, use the tip of a pencil to create a large area of the slice, ensuring the chromosomes are well dispersed for easy observation.

### Karyotype analysis

At least 30 metaphase cells with clear chromosomal morphology were selected to determine the location of centromere, and karyotype was analyzed according to their size and morphological characteristics. Relative chromosome length, arm ratio, and chromosome type were calculated using Levan's nomenclature (*Levan, Fredga & Sandberg, 2009*). In this classification system, chromosomes are categorized based on their relative lengths: those with a relative length of less than 2:1 are designated as Type A, while chromosomes exhibiting a relative length between 2:1 and 4:1 fall under Type B. Additionally, chromosomes that possess a relative length greater than 4:1 are classified as Type C. The coefficient of karyotype asymmetry can be classified according to the Stebbins standard (*Wang et al., 2020*).

**Table 1 Origin of materials.**

| Exp. Mat. Name | Sampling site | Long./E | Dim./N | Alt./m |
|---|---|---|---|---|
| MNH | Greenhouse of Guiyang University | 106.78 | 26.55 | 1,091 |
| KS09 | Xiaba town, Guiyang City | 106.80 | 26.51 | 1,250 |
| WD2 | Wudang District, Guiyang City | 106.92 | 26.71 | 1,242 |
| WD3 | Wudang District, Guiyang City | 106.92 | 26.71 | 1,242 |
| FHO2 | Xiaba town, Guiyang City | 106.80 | 26.51 | 1,250 |
| ZZ11 | Zhongzhai Town, Liuzhi Special District | 105.22 | 26.17 | 1,315 |
| XCX25 | Xinchang town, Liuzhi Special District | 105.31 | 26.43 | 1,433 |
| DCZC27 | Zhuhai Town, Panzhou City | 104.73 | 25.64 | 1,835 |
| CZC28 | Zhuhai Town, Panzhou City | 104.73 | 25.65 | 1,856 |
| LJCC29 | Jichang Ping town, Panzhou City | 104.63 | 25.89 | 1,854 |
| MT122 | Xinglong town, Meitan County | 107.56 | 27.71 | 824.69 |
| MT123 | Xinglong town, Meitan County | 107.56 | 27.71 | 824.69 |
| MT124 | Xinglong town, Meitan County | 107.56 | 27.71 | 824.69 |
| YQ129 | Dawujiang Town, Yuqing County | 107.61 | 27.63 | 880.85 |
| YQ130 | Songyan Town, Yuqing County | 107.62 | 27.56 | 839.34 |
| YQ131 | Ao Xi Town, Yuqing County | 107.65 | 27.46 | 764.07 |
| HZ139 | Anjia Village, Hezhang County | 104.49 | 27.24 | 2,476.37 |
| HZ144 | Zhu Ming Town, Hezhang County | 104.55 | 27.21 | 1,759.02 |
| HZ145 | Nimaogou Village, Hezhang County | 104.56 | 27.22 | 1,916.7 |
| HZ152 | Mustangchuan town, Hezhang County | 104.82 | 27.12 | 1,555.06 |
| RHC157 | Renhe Village, Qianxi City | 106.05 | 27.17 | 1,097.7 |
| RHC158 | Renhe Village, Qianxi City | 106.05 | 27.17 | 1,097.7 |
| CPC159 | Chunping Village, Qianxi City | 106.04 | 27.16 | 1,097.7 |
| WJZ5 | Dongfeng Village, Liuzhi Special District | 105.42 | 26.22 | 1,384 |
| ZZ12 | Zhongzhai Town, Liuzhi Special District | 105.22 | 26.17 | 1,315 |
| LCC16 | Longhe town, Liuzhi Special District | 105.43 | 26.30 | 1,171 |
| XCX23 | Xinchang town, Liuzhi Special District | 105.31 | 26.43 | 1,451 |
| DJPC31 | Liuguan Town, Panzhou City | 104.64 | 25.87 | 1,736 |

Note:
Exp. Mat. Name, Experimental material name; Long., longitude; Dim., dimensionality; Exp. Mat. Name, The experimental materials were labeled with Chinese pinyin initials representing the collection sites of Chinese cherry germplasm resources in different areas of Guizhou.

## Karyotype cluster analysis

In accordance with the cluster analysis methodology delineated by *Qiao, Wang & Cao (2020)*, a comprehensive systematic cluster analysis was executed utilizing SPSS software (IBM, Armonk, NY, USA). This analysis was systematically grounded in an array of eight meticulously selected karyotype parameters, which encompassed the longest to shortest chromosome ratio, the range of relative lengths, the average relative length, the average arm ratio, the relative length variation, the variation in arm ratios, the incidence of arm ratios exceeding a threshold of 2:1, as well as the karyotype asymmetry coefficient.

## Flow cytometry

Take 0.2 g of fresh young cherry leaves, gently rinse off any dust, wipe away any water stains with filter paper, and place them on a clean, chilled petri dish. Take 500 ul of nuclear lysate from the CyStain UV Precise P kit, add it around the sample, swiftly mince to a froth using a clean, sharp blade for complete nucleus extraction within 60 s. The liquid from the petri dish was filtered through a 50 um celltrics filter into a 1.5 mL centrifuge tube. A total of 2,000 ul of DAPI fluorescent dye solution from the CyStain UV Precise P kit was added to the tube and left to stain in darkness for 2 min. Through computer analysis, the nuclear DNA content of all test samples was compared to the control standard. Each sample was run in triplicate to minimize errors, with a minimum of 10,000 nuclei collected per test. In the control group, fluorescence signal intensity was compared to ascertain ploidy.

# RESULTS

## Identification of the ploidy levels of cherries

Cytogenetic analysis of 28 cherry accessions revealed a stable chromosome count with a base number of 8, comprising both tetraploid and hexaploid types (Figs. 1, 2, S1). Tetraploids, accounting for 18 accessions, exhibited four distinct karyotype formulas, including $2n = 4X = 32 = 32m$, $2n = 4X = 32 = 28m + 4sm$, $2n = 4X = 32 = 24m + 8sm$, and $2n = 4X = 32 = 20m + 12sm$. The remaining nine hexaploid accessions displayed two karyotype formulas: $2n = 6X = 48 = 48m$ and $2n = 6X = 48 = 42m + 6sm$. Notably, no pentaploids were detected. The karyotype formula was dominated by two types: metacentric (m) and sub metacentric (sm). Various accessions demonstrated different centromere configurations, with some featuring one, two, or three sets of mesocentric chromosomes, while others exhibited only the mesocentric type.

This study also investigated the ploidy levels of 14 cherry accessions from Guizhou Province using flow cytometry with the tetraploid accession MNH as an internal control (Fig. S2). Ten accessions (WD3, ZZ12, XCX23, DCZC27, CZC28, LJCC29, DJPC31, HZ144, HZ145) exhibited G1 peak values similar to MNH, indicating a tetraploid status. In contrast, four accessions (WD2, WJZ5, ZZ11, XCX25) exhibited G1 peak values of 152.97, 151.70, 151.46, and 156.89 relative fluorescence units (RFU), respectively. These values are approximately 1.5 times greater than that of the MNH accession, which recorded a G1 peak value of 103.15 RFU. This significant difference in G1 peak values suggests that the four accessions may possess a hexaploid genetic status.

## Karyotype characteristics of cherry

The analysis of chromosome morphology was conducted on 28 metaphase cells exhibiting distinct karyotypic characteristics. As presented in Table 2, the karyotype formula predominantly comprised two types of metacentric (m) chromosomes and a few sub metacentric (sm) chromosomes. Specifically, MNH, RHC157, YQ130, YQ131, and LCC16 exhibited one set of mesocentric chromosomes, while HZ152 had two sets, and ZZ12 had three sets. The remaining Chinese cherry accessions displayed the middle centromere type. Significant variability was observed among the 28 Chinese cherry accessions in terms of karyotypes parameter. The mean arm ratio (MAR) ranged from 1.15 to 1.56, and the

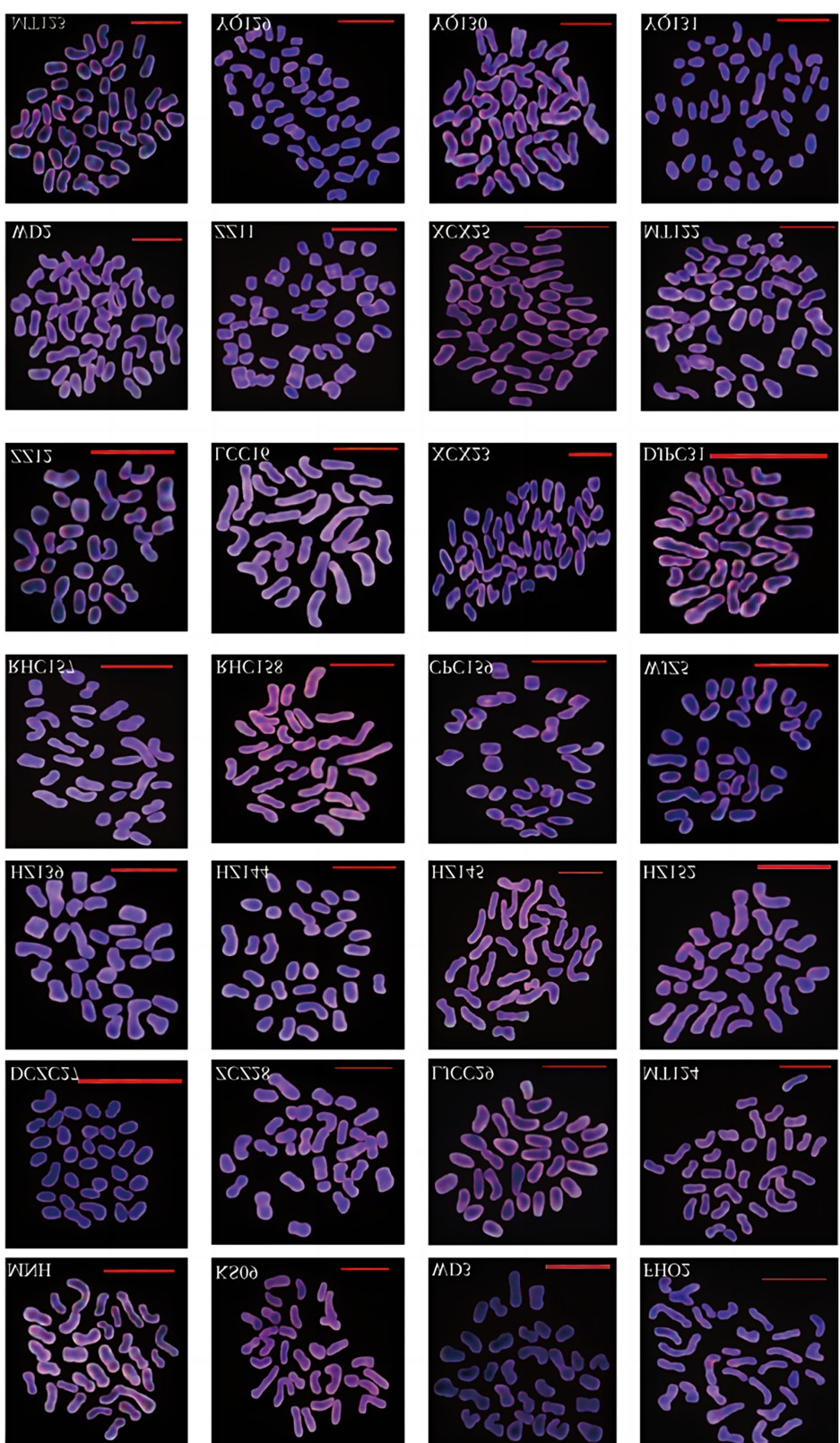

**Figure 1 Chromosomes of 28 cherry accessions from Guizhou Province during metaphase cleavage phase.**

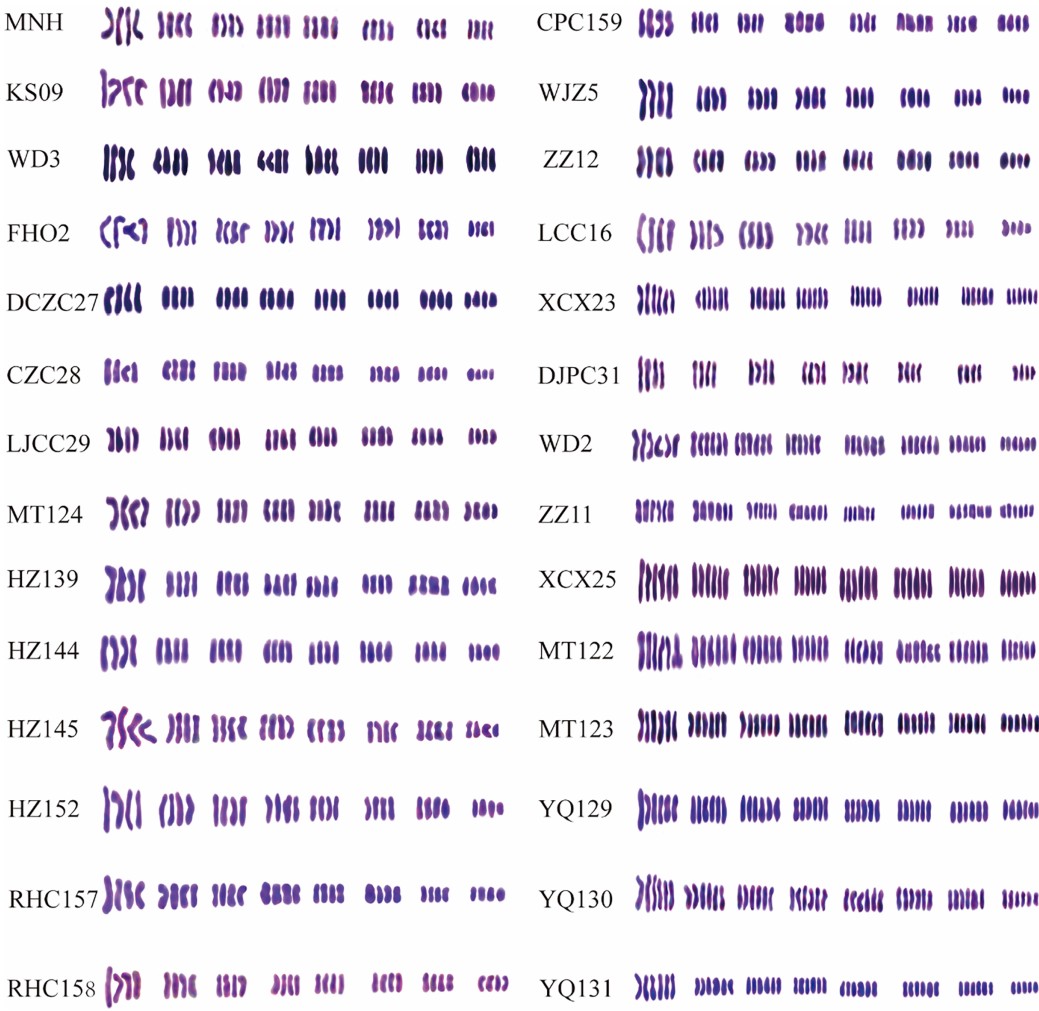

**Figure 2 Chromosomal karyotype in 28 cherry accessions from Guizhou Province.**

longest chromosome/shortest chromosome (Lc/Sc) varied among the 15 accessions, including MNH, KS09, WD2, FHO2, CZC28, MT124, YQ130, YQ131, HZ139, HZ144, HZ145, HZ152, RHC157, ZZ12, and XCX23. The index of the karyotypic asymmetry (As. K) ranged from 53.74% to 61.60%. Three distinct karyotype types were identified: five accessions were classified as type 2B (MNH, YQ130, HZ152, RHC157, and ZZ12), 10 were type 1B (KS09, WD2, FHO2, CZC28, MT124, YQ131, HZ139, HZ144, HZ145, and XCX23), and 13 were type 1A (WD3, ZZ11, XCX25, DCZC27, LJCC29, MT122, MT123, YQ129, RHC158, CPC15, WJZ5, LCC16, and DJPC31).

In the context of phylogenetic evolution, plants typically exhibit relatively symmetrical karyotypes, showcasing a consistent pattern in their nuclear structure. By utilizing the mean arm ratio and chromosome length ratio as axes, a scatter plot can be generated to visualize the degree of karyotype asymmetry, thereby reflecting the evolutionary stage of the germplasm (Fig. 3). The descending order of nuclear asymmetry degree among these accessions as follows: HZ152 > ZZ12 > MNH > YQ130 > RHC157 > LCC16 > RHC158 >

**Table 2  Karyotypes parameter of the tested cherry accessions from Guizhou Province.**

| Accessions | Karyotype formula | MAR | Lc/Sc | MCI | As.K(%) | karyotype |
|---|---|---|---|---|---|---|
| MNH | 2n = 4X = 32 = 28m + 4sm | 1.55 | 2.01 | 0.40 | 61.09 | 2B |
| KS09 | 2n = 4X = 32 = 32m | 1.24 | 2.16 | 0.45 | 54.9 | 1B |
| WD2 | 2n = 6X = 48 = 48m | 1.35 | 2.14 | 0.43 | 57.71 | 1B |
| WD3 | 2n = 4X = 32 = 32m | 1.23 | 1.68 | 0.45 | 55.11 | 1A |
| FHO2 | 2n = 4X = 32 = 32m | 1.32 | 2.4 | 0.43 | 56.96 | 1B |
| ZZ11 | 2n = 6X = 48 = 48m | 1.30 | 1.64 | 0.43 | 56.56 | 1A |
| XCX25 | 2n = 6X = 48 = 48m | 1.41 | 1.87 | 0.42 | 58.75 | 1A |
| DCZC27 | 2n = 4X = 32 = 32m | 1.15 | 1.76 | 0.47 | 53.74 | 1A |
| CZC28 | 2n = 4X = 32 = 32m | 1.24 | 2.00 | 0.45 | 55.48 | 1B |
| LJCC29 | 2n = 4X = 32 = 32m | 1.25 | 1.60 | 0.45 | 55.62 | 1A |
| MT122 | 2n = 6X = 48 = 48m | 1.38 | 1.89 | 0.42 | 58.31 | 1A |
| MT123 | 2n = 6X = 48 = 48m | 1.29 | 1.75 | 0.44 | 56.43 | 1A |
| MT124 | 2n = 4X = 32 = 32m | 1.31 | 2.04 | 0.44 | 56.85 | 1B |
| YQ129 | 2n = 6X = 48 = 48m | 1.24 | 1.73 | 0.45 | 55.49 | 1A |
| YQ130 | 2n = 6X = 48 = 42m + 6sm | 1.46 | 2.06 | 0.41 | 59.73 | 2B |
| YQ131 | 2n = 6X = 48 = 42m + 6sm | 1.37 | 2.22 | 0.42 | 58.03 | 1B |
| HZ139 | 2n = 4X = 32 = 32m | 1.26 | 2.01 | 0.44 | 55.68 | 1B |
| HZ144 | 2n = 4X = 32 = 32m | 1.42 | 2.05 | 0.41 | 58.63 | 1B |
| HZ145 | 2n = 4X = 32 = 32m | 1.43 | 2.44 | 0.41 | 58.62 | 1B |
| HZ152 | 2n = 4X = 32 = 24m + 8sm | 1.56 | 2.65 | 0.40 | 61.60 | 2B |
| RHC157 | 2n = 4X = 32 = 28m + 4sm | 1.44 | 2.21 | 0.42 | 59.30 | 2B |
| RHC158 | 2n = 4X = 32 = 32m | 1.43 | 1.82 | 0.41 | 59.02 | 1A |
| CPC159 | 2n = 4X = 32 = 32m | 1.28 | 1.79 | 0.44 | 56.16 | 1A |
| WJZ5 | 2n = 6X = 48 = 48m | 1.20 | 1.75 | 0.45 | 54.79 | 1A |
| ZZ12 | 2n = 4X = 32 = 20m + 12sm | 1.55 | 2.45 | 0.40 | 61.41 | 2B |
| LCC16 | 2n = 4X = 32 = 28m + 4sm | 1.43 | 1.97 | 0.41 | 59.09 | 1A |
| XCX23 | 2n = 4X = 32 = 32m | 1.27 | 2.54 | 0.44 | 55.79 | 1B |
| DJPC31 | 2n = 4X = 32 = 32m | 1.24 | 1.91 | 0.45 | 55.70 | 1A |

Note:
  MCI, mean centromeric index; As.K, index of the karyotypic asymmetry; Lc/Sc, longest chromosome/shortest chromosome; MAR, mean arm ratio (arm ratio = length of the long arm/length of the short arm).

XCX25 > HZ144 > HZ145 > YQ131 > WD2 > FHO2 > MT124 > ZZ11 > MT123 > CPC159 > XCX23 > DJ PC31 > HZ139 > LJCC29 > YQ129 > CZC28 > WD3 > KS09 > WJZ5 > DCZC27.

## Cluster analysis of cherry in Guizhou Province

The genetic diversity of 28 cherry accessions originating from Guizhou Province was assessed through a cluster analysis grounded in chromosomal parameters. As illustrated in Fig. 4, these 28 accessions were effectively categorized into six distinct groups according to branch counts, utilizing a similarity coefficient threshold of 0.5. The first group consisted of WD3, YQ129, WJZ5, LJCC29, ZZ11, CPC159, DJPC31, MT123, and DCZC27, while the second group included XCX25, MT122, RHC158, LCC16, HZ144, WD2, MT124, KS09,

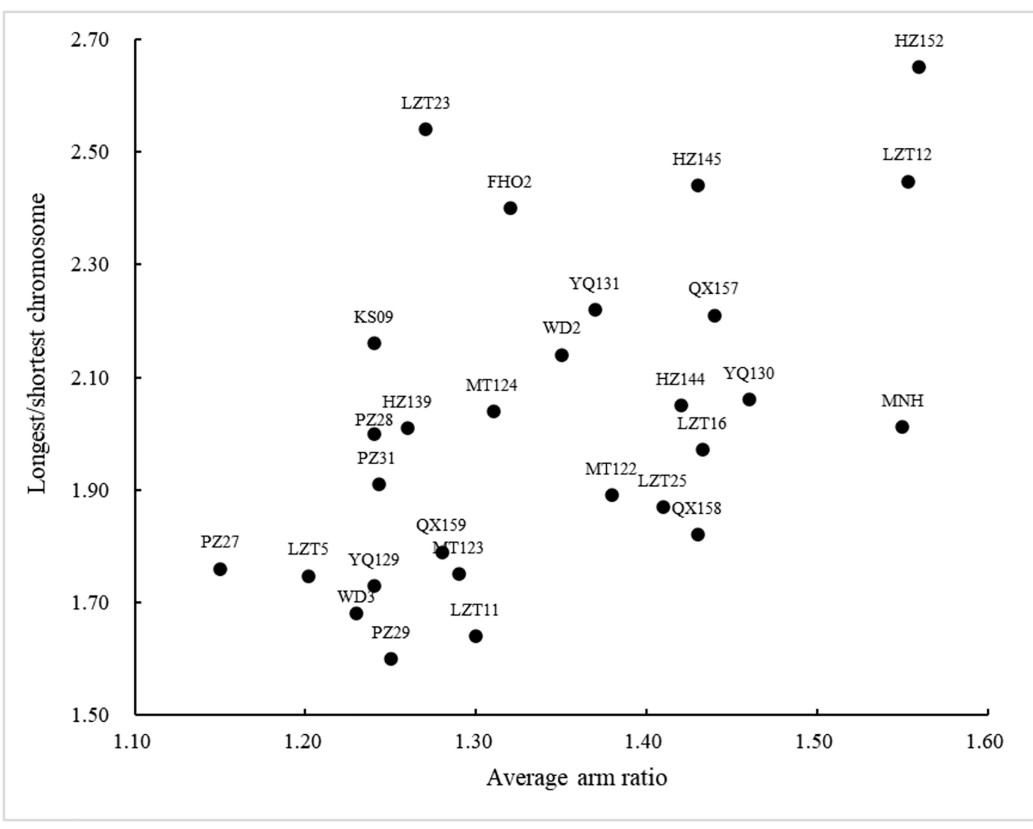

**Figure 3  Scatter plot of karyotypic asymmetry.**

HZ139, and CZC28. The third, fourth, fifth, and sixth groups were composed of FHO2, XCX23, and HZ145; HZ152 and ZZ12; MNH, YQ130, and RHC157; and YQ131, respectively. Furthermore, HZ152, ZZ12, MNH, YQ130, and RHC157 exhibited relatively low similarities compared to the other 23 Chinese cherry accessions. This analysis revealed notable genetic differentiation among the examined germplasm resources, indicating the presence of valuable genetic diversity within the cherry population from this region.

## DISCUSSION

The present study examined the karyotypic variations among 28 native cherry accessions from Guizhou Province. The karyotype formula, Lc/Sc, MAR, and As.K were analyzed to provide insights into the structural composition of the genome. The results indicated that the cherry accessions exhibited a relatively symmetric karyotype, with the As.K values ranging from 53.74% to 61.6%, suggesting a low degree of evolution. This finding was consistent with the previous report on the As.K of Auriculata species (59.41%, *Zhao, Li & Gao, 2014*). Interestingly, despite the similar ploidy, Chinese cherry cultivars showed diverse karyotype formulas, a phenomenon also observed in passionflower plants (*Qian et al., 2023*). The assessment of chromosome parameters, particularly the calculation of As.K (asymmetry index), is increasingly recognized as a crucial tool in elucidating genetic relationships and evolutionary patterns among various plant species. This analysis facilitates a detailed understanding of karyotype structure and variation, offering insights

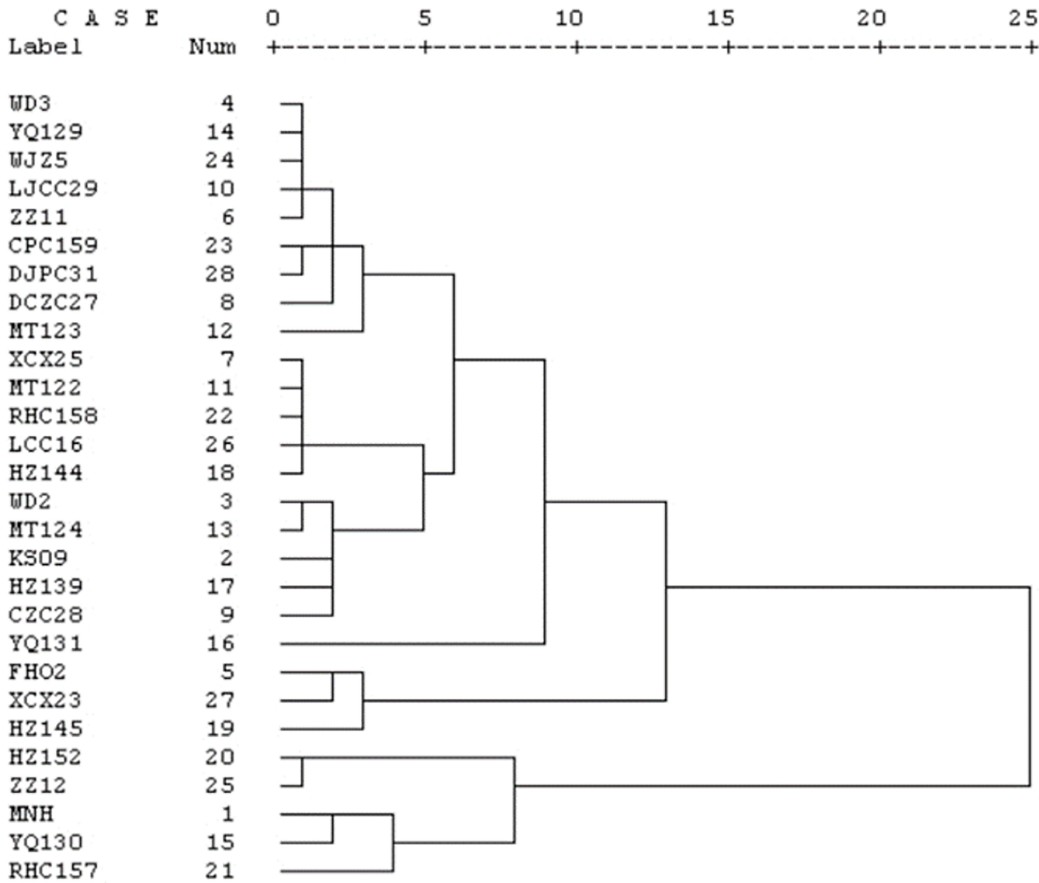

**Figure 4 Karyotypic clustering dendrogram.** Label represented 28 cherry accessions used in this study, Num represented genetic distance.

into the adaptive strategies and evolutionary trajectories of distinct taxa. Research has demonstrated that karyotype analysis enables the classification of plant karyotypes, aiding in the identification of evolutionary dynamics both within species and across phylogenetic groups (*Sun et al., 2019*; *Seijo & Fernández, 2003*). For instance, studies have shown that specific karyotypic features can correlate with ecological and morphological diversity, thereby suggesting a meaningful link between chromosomal architecture and evolutionary processes. Therefore, the integration of As.K assessments with traditional morphological and molecular approaches can provide a more comprehensive understanding of plant biodiversity and evolutionary history (*Hu et al., 2015*).

The identification of the karyotypes "1A", "1B", and "2B" within the studied materials significantly contributes to our understanding of the genetic diversity present in the genus Prunus, particularly within the context of Chinese germplasm. Notably, karyotype HZ152 emerges as the most advanced evolutionary combination, indicative of potential adaptive advantages, such as improved stress resilience or enhanced fruit quality, while karyotype DCZC27 represents a more primitive form. This juxtaposition underscores the importance of embracing a wide and novel breeding perspective that encompasses not only traditional selection methods but also contemporary genomic techniques. By integrating these

findings into breeding programs, researchers can potentially harness the genetic traits associated with various karyotypes, fostering the development of new cherry cultivars that are better suited to the changing climate and market demands in China. Additionally, a comprehensive review of recent literature reveals a notable distribution of karyotypes among cerasus and cherry germplasm throughout China, highlighting significant regional variations that reflect both historical agricultural practices and ecological factors (*Qian et al., 2023*; *Sun et al., 2019*; *Wang et al., 2018*). This distribution, characterized by local adaptations and the presence of unique genetic traits, underscores the critical need for conservation strategies and informed breeding programs that leverage the rich genetic resources found in Chinese cherry cultivars. Thus, understanding the karyotype distribution offers valuable insights for breeders seeking to optimize genetic diversity and resilience in cherry production across diverse environmental contexts.

The findings reveal that among the 28 accessions evaluated, a chromosomal base of $x = 8$ was established, comprising 19 individuals classified as tetraploid and nine as hexaploid. This distribution of ploidy levels is noteworthy, as it indicates a diverse genomic foundation that may significantly impact both the horticultural traits and adaptability of these cherry varieties. The dominance of tetraploid individuals is consistent with existing literature suggesting that tetraploidy can enhance fruit quality and yield (*Wang et al., 2018*). Furthermore, the presence of hexaploid individuals invites intriguing inquiries into their potential resilience against environmental stressors, which is vital for breeding programs focused on improving cherry cultivation in China. Overall, these findings enhance our understanding of the genetic diversity within Chinese cherries and highlight the need for further research into their ploidy levels and associated phenotypic characteristics.

The cluster analysis results revealed that when the similarity coefficient exceeded 5, the 28 cherry accessions from Guizhou Province could be classified into six distinct groups. Interestingly, cherry accessions from the same region did not cluster together, indicating no apparent relationship between the karyotype of the cherry accessions and their geographic origin. Furthermore, the study found no consistent correlation between the chromosome number and the same species. Scatter plots and cluster analysis demonstrated considerable diversity among the different accessions, which may be attributed to geographical variations, elevation differences, or climatic conditions. In a related study, *Li et al. (2021)* employed karyotype similarity coefficients to determine the karyotype asymmetry and perform cluster analysis on Epimedium species. Notably, the majority of the 28 cherry accessions examined in the current experiment belonged to the 1A karyotype. The observed differences in chromosome karyotypes could be linked to factors such as chromosome size, the position of the centromere, and the degree of chromosome coiling. Additionally, no satellite structures were identified in this study, which may be due to the relatively small chromosome size or the challenges in identification and observation resulting from excessive chromosome condensation. Hence, the optimization of the chromosome preparation technique for Chinese cherry warrants further investigation. The 28 cherry accessions examined exhibited a diverse range of chromosome types and karyotypes, which may be attributed to variations in compression methods or data measurement. Additionally, differences in the experimental materials, their sources, and

the processing reagents and durations used in each step of the chromosome preparation process could potentially influence the observed results. These factors should be carefully considered and standardized to ensure the reliability and reproducibility of the karyotype analyses. Addressing these methodological aspects will contribute to a more comprehensive understanding of the cytogenetic characteristics of Chinese cherry.

This study determined the ploidy levels of 28 Chinese cherry cultivars in Guizhou using chromosome counting and flow cytometry. The results revealed that most cultivars were tetraploid, with a small proportion being hexaploid. Notably, four distinct tetraploid karyotype formulas were identified, ranging from $2n = 4X = 32 = 32m$ to $2n = 4X = 32 = 20m + 12sm$. These findings contribute to the understanding of ploidy variation within the *Prunus pseudocerasus* Lindl., which is known to exhibit diverse karyotypes. Previous research has documented various karyotypes in related species, such as *Bletilla* (*He et al., 2022*), *Pseudocerasus* (*Jiu et al., 2024*), and other Chinese cherry varieties (*Wang et al., 2022*). This study's accurate chromosome identification of Chinese cherry cultivars provides novel insights into the karyotype diversity within the *Prunus pseudocerasus* Lindl. and complements existing knowledge about the complex relationships between diploid and polyploid cherry species (*Tavaud et al., 2004*).

## CONCLUSION

The combined use of flow cytometry and chromosome compression analysis revealed that the 28 cherry accessions examined exhibited a chromosomal base of $x = 8$, comprising 19 tetraploid and nine hexaploid individuals. These results contribute to a more accurate identification of ploidy levels in Chinese cherry species, which is essential for understanding genetic relationships, identifying desirable traits, and facilitating the breeding of superior cherry varieties. The cytological techniques employed in this study offer important technical support for ongoing research in the field of Chinese cherry genetics and breeding.

### Funding

This study was supported by Special Project for Cultivating Academic New Talents and Free Exploration and Innovation of the Department of Science and Technology of Guizhou Province, Forestry scientific research project of Guizhou Provincial Forestry Bureau (QLKH[2020]08), the program for Natural Science Research in Guizhou Education Department (QJJ-[2023]-024). The funders had no role in study design, data collection and analysis, decision to publish, or preparation of the manuscript.

### Grant Disclosures

The following grant information was disclosed by the authors:
Department of Science and Technology of Guizhou Province.
Forestry scientific Research Project of Guizhou Provincial Forestry Bureau: QLKH[2020]08.

Program for Natural Science Research in Guizhou Education Department: QJJ-[2023]-024.

## Competing Interests

The authors declare that they have no competing interests.

## Author Contributions

- Nian Chen performed the experiments, analyzed the data, prepared figures and/or tables, authored or reviewed drafts of the article, and approved the final draft.
- Yali Wang performed the experiments, analyzed the data, prepared figures and/or tables, and approved the final draft.
- Mei He analyzed the data, authored or reviewed drafts of the article, and approved the final draft.
- Fei An performed the experiments, authored or reviewed drafts of the article, and approved the final draft.
- Jiyue Wang analyzed the data, authored or reviewed drafts of the article, and approved the final draft.
- Changmei Song conceived and designed the experiments, authored or reviewed drafts of the article, and approved the final draft.

## Data Availability

The data are available in the Supplemental Files.

## Supplemental Information

Supplemental information for this article can be found online at http://dx.doi.org/10.7717/peerj.18668#supplemental-information.

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
