# Peer review of "Identification of chromosome ploidy and karyotype analysis of cherries (Prunus pseudocerasus Lindl.) in Guizhou"

_PeerJ, doi:10.7717/peerj.18668_

## Round 0.1 · original submission · Major Revisions

Please address the concerns raised by the reviewers. Specifically, the comparisons between the existing literature and the current work are crucial.

Reviewer 1 ·

Basic reporting

No comment

Experimental design

Add method of Karyotype diagram

Validity of the findings

no comment

Additional comments

It is clear and add some comment in "Add method of Karyotype diagram"

Annotated reviews are not available for download in order to protect the identity of reviewers who chose to remain anonymous.

·

Basic reporting

General Explanations
This study is an original research article with figures and tables and contains valuable information for Cerasus genus using recent findings.
English is enough for academic languages.

Experimental design

The current report is suitable according to scope. This research stated clearly how research fills an identified knowledge gap. However, there were several points to be corrected and edited. I have detailed in below.

Validity of the findings

It has novel findings and valuable results.

Additional comments

The current paper is related the “Identification of chromosome ploidy and karyotype analysis of cherries (Prunus pseudocerasus Lindl.) in Guizhou”.
General Explanations
This study is an original research article and contains valuable information for Cerasus genus and its species.
The introduction generally included the germplasm of the Cerasus in China and karyotype analysis technics. Although the aims were clearly explained in the Introduction, insufficient points need to be revealed such as the importance of the ploidy levels in cross-breeding, dis/advantages challenges over the breeding to facilitate and mitigate breeding processes using recent findings. Since different of the ploidy degree of the plants can be reasoned to deadly results in breeding due to incompatibility of the sister chromatid during the crossover. And, what kind of karyotypes based on ploidy levels can affect the genetic architectures between genus and species? This should be explained to increase the present paper's quality and to understand the genetic diversity and domestication within the genus.
In Mat met section, cluster analysis methods should be expressed in detail parameters. There was no information about it. And, A and B karyotypes should be defined to determine or decide A or B, which should be in Mat Met.
In Results section,
Line 103, subtitle should be “identification of the ploidy levels of cherries”
Line 108-109, mesocentric (m) and submetacentric (sm) expressions should be in Mat Met.
Line 115, “G1 peak” should be detailed in mat met into related section.
Line 140, please remove “Guizhou province” words from subtitle and edit title.
In cluster analysis, results can be detailed with dendrogram results on the branches counts.
In discussion section,
There were a narrow discussion field in this part. In the literature, there were several related reports performed in Cerasus and sweet cherries. However, this did not discuss. Please, add a combined discussion with these findings using a wide and novel breeding perspectives. And, a paragraph needs to discussion the distribution of the karyotypes of cerasus and cherries in China germplasm using recent articles. Also, ploidy levels should be discussed, since there was no information although this study is associated with ploidy detection.
Line 160-164, this part looks like to introduction information, please remove them or discuss with more detail and necessary values.
In figures, please use Arial characters during the printing. And, the figure legends should be contained more correct information. That’s why, please remove “Guizhou province” from legends.
Especially in figure 3, reprint to see the counts on figures.

I suggest that this manuscript can accept for publication after major revision.

·

Basic reporting

This work presented by Chen et al., is novel and provides new contributions to the knowledge of the cytogenetics of such an important species as Prunus pseudocerasus Lindl. This work is well structured, and the language used is correct. The figures are also correct and reflect properly the results obtained. However, there are some aspects that I would like the authors to consider:

• The first paragraph of the introduction (lines 37 to 44) does not correctly allude to the taxonomy of the genus Prunus L. Cerasus is a subgenus of the genus Prunus (Shi et al., 2013), so Cerasus as a genus name is not valid. Thus, it is not correct to say “The genus Cerasus” or “Cerasus pseudocerasus (Lindl.) G. Don”, as they are not valid names. Instead, it would be more correct to say, “The subgenus Cerasus” and “Prunus pseudocerasus Lindl.” respectively. The authors should check this error, as it appears elsewhere in the manuscript (eg. lines 56, 188, 192). Also, I do not know exactly what the authors mean by the name Pseudoprunus, as there does not seem to exist such a name for a genus or any taxonomic category or cultivar variety.

• In the final paragraph of the Introduction (lines 61 to 68), the authors declare that the ploidy level of the Chinese cherry (Prunus pseudocerasus Lindl.) remain largely unknown, but I have found that there are several works (Wang et al., 2018, Xueou et al., 2018, Wang et al., 2023) that delve into this topic. In this way, I would like the authors to develop this issue in this paragraph.

• In section “Chromosome preparation” (line 72) and “Flow cytometry” (line 92) of Materials and Methods, the processes that the authors have followed are described in detail, but no work is cited where these methods are described. Have the authors followed a method that has already been described? If so, it should be cited. In addition, I do not find in Materials and Methods any paragraph where the authors describe how they have performed the karyotypic clustering dendrogram.

• In the section “Karyotype characteristics of cherry in Guizhou province” (lines 127 to 131), in Results, the different accessions are classified in three different Karyotype types (1A, 1B and 2B), but the authors don’t reference the work where these karyotype types are firstly stablished (Wang et al., 2018).

• In the second paragraph of section “Karyotype characteristics of cherry in Guizhou province” (lines 132 to 139), in Results, the authors declare that “plants in early or primitive stages typically exhibit relatively symmetrical karyotypes, showcasing a consistent pattern in their nuclear structure.”, ¿could the authors cite a work where this is demonstrated? Also in this paragraph, the expressions “most evolved” or “most primitive” are not correct, since there are no organisms “more evolved” than others, but more or less organized or differentiated. This expression is incorrect here also because the authors have not done any analysis to test the phylogenetic relationships between the different Karyotypes. Also check this in the manuscript, as it appears elsewhere (eg. Lines 155 and 159).

• In the Discussion, in lines 188 to 189, Have the authors read any work to say that the species of the genus Cerasus (which is an invalid name for a genus, it should be Prunus) display different karyotypes? If so, it should be referenced.

• Also in the discussion, I would like the authors to compare their results with the results obtained by previous works (Wang et al., 2018, Xueou et al., 2018, Wang et al., 2023).

• I would like the authors to better explain figure 6 in the legend, detailing the meaning of “Label” and “Num”, as well as the numerical scale above the figure.

Experimental design

The Experimental design of this work is the appropriate for the objectives proposed. I would like to point that the authors have conduct an extensive sampling of Prunus pseudocerasus in the Guizhou Province. The methods are described properly, but as mentioned above, I would like the authors to cite the works or the sources in which these methods were described firstly. Also, there is data available in other works (Wang et al., 2018, Xueou et al., 2018, Wang et al., 2023) that the authors could use in their analysis to compare the karyotype configuration of its samples from Guizhou Province with other samples from other Provinces of China. I believe that if these samples were included in the analysis of karyotypic asymmetry and Karyotypic clustering dendrogram, they would enrich the results of the work.

Validity of the findings

The work of Chen et al. is novel, as there are no previous karyotypic studies of the Chinese cherry in Guizhou Province, a species of great relevance for its utilization, in China and in many other parts of the world. However, there are very similar karyotypic studies of this species in other Provinces of China (Wang et al., 2018, Xueou et al., 2018, Wang et al., 2023), but some of these works are not cited in this work. From what I have had the opportunity to see in these works, the results obtained by the authors do not seem to be very different from those already obtained in other provinces of China, so I would like the authors to include these samples in some of their analyses (see previous section).

Additional comments

The present work is original and novel, as well as well structured. In addition, the language used is appropriate, making the content perfectly understandable. Nevertheless, while waiting for the authors to include in their analyses the data from the articles referenced in this review, I propose the status of this manuscript as Major Revision. In addition, I would like the authors to conduct a more comprehensive literature search for articles related to the present study. I believe that both aspects could enrich the content and results of the present work.

---

## Round 0.2 · Minor Revisions

Please address the specific remaining concerns raised by the reviewer.

·

Basic reporting

The manuscript provided by Chen et al. has been considerably improved after revision. The introduction is now more complete, as are the materials and methods. Also, the discussion is now better justified with the new references incorporated. In addition, the inclusion of Mei Hei as author is well justified in the rebuttal letter. There are only two minor aspects that I would like to note:
• In line 38 it is incorrect to say that Prunus pseudocerasus is a subgenus, as it is a species. This must be corrected.
• In Figure 4 I would like the authors to include in the legend the explanation that they have given me in the response letter about the meaning of “Label”, “Num” and the numerical scale to a better understanding of the figure.

Experimental design

No comments

Validity of the findings

No comments

Additional comments

None

---

## Round 0.3 · accepted · Accept

All concerns have been addressed. I would like to thank the authors for all their efforts.